# Addition of Chromosome 17 Polysomy and *HER2* Amplification Status Improves the Accuracy of Clinicopathological Factor-Based Progression Risk Stratification and Tumor Grading of Non-Muscle-Invasive Bladder Cancer

**DOI:** 10.3390/cancers14194570

**Published:** 2022-09-21

**Authors:** Ildikó Kocsmár, Éva Kocsmár, Gábor Pajor, Janina Kulka, Eszter Székely, Glen Kristiansen, Oliver Schilling, Péter Nyirády, András Kiss, Zsuzsa Schaff, Péter Riesz, Gábor Lotz

**Affiliations:** 1Department of Pathology, Forensic and Insurance Medicine, Semmelweis University, Üllői Street 93, H-1091 Budapest, Hungary; 2Department of Urology, Semmelweis University, Üllői Street 78b, H-1082 Budapest, Hungary; 3Department of Pathology, Medical School and Clinical Center, University of Pécs, Szigeti Street 12, H-7624 Pécs, Hungary; 4Department of Pathology, University Hospital Bonn, Universitätsklinikum Bonn (AöR), Venusberg-Campus 1 Building 62, 53127 Bonn, Germany; 5Institute of Surgical Pathology, Medical Center, University of Freiburg, Breisacher Street 115A, 79106 Freiburg im Breisgau, Germany

**Keywords:** Chromosome 17 polysomy, *HER2* amplification, non-muscle-invasive bladder cancer, progression risk stratification, p53 expression, fluorescence in situ hybridization

## Abstract

**Simple Summary:**

The addition of chromosome 17 polysomy/*HER2* amplification status to the updated EAU and AUA scores improves their accuracy, allowing molecular reclassification of EAU high-risk NMIBCs and G2 tumors. Based on this, we propose to reclassify the non-*HER2* amplified, non-polysomic EAU 3–4 (high- and very high-risk) cases to the EAU 2 (intermediate) risk group to prevent unnecessarily strict follow-up and treatment for these patients. Furthermore, to classify Chr17 polysomic and/or *HER2* amplified G2 tumors as high-grade (HG) and non-*HER2* amplified, non-polysomic G2 tumors as low-grade (LG) NMIBCs (G1 and G3 tumors remain graded as low- and high-grade, respectively). Thus, the implementation of Chr17 polysomy/*HER2* amplification testing would provide an immediate and simple solution to further refine the prognostic risk assessment of NMIBCs in the uro-oncology practice.

**Abstract:**

Progression of non-muscle-invasive bladder cancer (NMIBC) to muscle-invasive disease (MIBC) significantly worsens life expectancy. Its risk can be assessed by clinicopathological factors according to international guidelines. However, additional molecular markers are needed to refine and improve the prediction. Therefore, in the present study, we aimed to predict the progression of NMIBCs to MIBC by assessing p53 expression, polysomy of chromosome 17 (Chr17) and *HER2* status in the tissue specimens of the tumors of 90 NMIBC patients. Median follow-up was 77 months (range 2–158). Patients with Chr17 polysomy or *HER2* gene amplification had a higher rate of disease progression (hazard ratio: 7.44; *p* < 0.001 and 4.04; *p* = 0.033, respectively; univariate Cox regression). Multivariable Cox regression models demonstrated that the addition of either Chr17 polysomy or *HER2* gene amplification status to the European Association of Urology (EAU) progression risk score increases the c-index (from 0.741/EAU/ to 0.793 and 0.755, respectively), indicating that Chr17 polysomy/*HER2* amplification status information improves the accuracy of the EAU risk table in predicting disease progression. *HER2*/Chr17 in situ hybridization can be used to select non-progressive cases not requiring strict follow-up, by reclassifying non-*HER2*-amplified, non-polysomic NMIBCs from the high- and very high-risk groups of EAU to the intermediate-risk group.

## 1. Introduction

Bladder cancer is the 10th most common type of cancer worldwide, which is the ninth leading cause of cancer death in men [1]. Approximately 75% of the cases are non-muscle invasive bladder cancer (NMIBC) at the time of diagnosis; however, 10–20% of them progress into the potentially life-threatening muscle-invasive bladder cancer (MIBC). Therefore, NMIBCs should be frequently monitored for many years, which also makes management of this disease economically demanding [2,3]. The European Association of Urology (EAU) and the American Urological Association (AUA) both propose clinicopathological factor-based stratification of patients into prognostic risk groups [4]. Currently, the EAU guidelines are the most frequently used ones by clinicians across Europe [5]. The original EAU progression risk groups were introduced in 2013 based on the European Organization for Research and Treatment of Cancer (EORTC) risk score, and have been recently updated [3,4,6]. The updated EAU prognostic risk stratification for NMIBC incorporates both the WHO 1973 and WHO 2004/2016 grading systems, providing two progression probability results for each case (EAU WHO 1973 and EAU WHO 2004/2016). However, further refinement of the risk progression scores is desirable by adding molecular markers, for which the results of various studies to date are promising but not yet suitable for routine clinical practice [2,6].

Bladder cancer progression is accompanied by increased chromosomal instability and aneuploidy [7,8]. Cytogenetic studies revealed frequent alterations of a variety of chromosomes including chromosome 17 (Chr17) in bladder cancer; they have also demonstrated that its high-polysomy in the urothelial cancer cells is associated with progression of NMIBCs into muscle invasive disease [7,8,9,10]. HER2 is a cell membrane surface-bound receptor tyrosine kinase, which is involved in signaling pathways leading to cell growth and differentiation [11]. It is encoded by *HER2/neu* proto-oncogene located on the q arm of Chr17 (17q21-22). Previous studies have shown that *HER2* amplification, and/or overexpression of its protein, is associated with poor prognosis in NMIBCs [12,13,14,15]. Beside *HER2* gene amplification, polysomy 17 can also lead to an elevated HER2 protein level due to gain of *HER2* gene [16,17]. As a tumorsuppressor gene located on the p arm of Chr17 (17p13), *TP53* gene and encoded p53 protein are responsible for the maintenance of the genomic integrity and therefore mutations in this gene, representing one of the most critical events in human carciogenesis [18]. Overexpression of the p53 protein correlates with the mutational status of *TP53* and is prognostically significant in high grade urothelial cancer [19,20].

Although the role of the molecular markers discussed above has been extensively investigated in the etiopathogenesis of bladder cancer and their prognostic role is well established, their practical value in improving the EAU and AUA prognostic classifications of non-muscle-invasive bladder cancer, and thus in predicting muscle invasiveness, has yet to be explored. Therefore, we aimed to investigate whether their detection by the widely available immunohistochemical and in situ hybridization assays used in routine pathological practice has added value to the well-known clinicopathological factors underlying the updated EAU and AUA progression risk groups. We also aimed to evaluate whether the prognostic value of the updated EAU risk stratification improved over the original EAU risk groups in our cohort.

Here, we present that polysomy 17 and *HER2* amplification status significantly improves the accuracy of the conventional tumor grading and the EAU progression risk stratification of NMIBC.

## 2. Materials and Methods 

### 2.1. Patient Selection and Data Collection

Tissue samples were retrospectively collected from 90 consecutive cases of NMIBC obtained by transurethral bladder tumor resection (TURBT), and 12 non-malignant cases as a control group (two samples with no pathological alteration, six with cystitis, two with urothelial hyperplasia and two with urothelial papilloma), from patients treated at the Urology Clinics of Semmelweis University and the University of Pécs between 2004–2006. NMIBC patients with variant histology, history of MIBC, malignant tumor of the upper urinary tract or other organ system or hematologic malignancies were excluded from the study. Of the 70 primary and 20 recurrent NMIBC patients, 42 were in follow-up for stage pTa disease, 47 for pT1, and 1 for pTis. The study was performed according to the Declaration of Helsinki and was approved by the Regional and the National Ethics Committees (#21/2019 and #14383-2/2017/EKU, respectively).

Variables collected from patients’ medical records included age, gender, tumor grade, stage, size, multiplicity, primary or recurrent nature, adjuvant instillation (BCG or chemo ever), follow-up time (in months), time-to-progression (in months) and recurrence. Staging and grading were re-evaluated by a uropathology expert (E.S.): cases were graded according to both 1973 (Grade 1–3) and 2004/2016 (low-grade/high-grade) World Health Organization (WHO) classifications for urothelial neoplasms, and stages were classified according to the 8th edition of AJCC/UICC TNM classification [21]. Patient follow-up was performed according to the national guidelines and was censored at the time of the most recent cystoscopy. Patients were evaluated for disease progression, defined as a recurrence when pathological examination confirmed muscle-invasive tumor (stage T2 or higher). Time-to-progression was measured from the time of TURBT for the tumor analyzed to the time of the progression event. Risk assessment of the NMIBCs was performed using both the WHO 2004/2016 and WHO 1973 classification systems for grade: samples were categorized into low-, intermediate-, high- and very high-risk groups for NMIBC, according to the original version and the 2021 update of the EAU prognostic factor risk groups [2,6]. Risk stratification was also performed based on the risk groups of the EORTC, and according to the current guideline of the AUA for non-muscle-invasive bladder cancer [3,22].

### 2.2. Immunohistochemistry (IHC) and Fluorescence In Situ Hybridization (FISH) Analysis

Tissue sections of 3–5 μm thickness were prepared from standard formalin-fixed, paraffin-embedded (FFPE) tissue blocks, and stained with hematoxylin-eosin (HE). HER2 IHC and FISH were performed according to the ASCO/CAP 2018 guidelines for breast cancer (Figure 1) [23]. Standard HER2 immunohistochemistry (IHC) was carried out with anti-c-erbB-2 antibody (Novocastra, Newcastle upon Tyne, UK; clone CB11), and p53 immunohistochemistry (IHC) was carried out with anti-p53 antibody (Dako, Glostrup, Denmark; clone DO-7) using Ventana Benchmark Ultra automated immunostainer and UltraView DAB detection kit (Ventana Medical Systems, Tucson, AZ, USA). Images were taken using Olympus BX51 microscope equipped with a DP70 color camera (Olympus, Tokyo, Japan). HER2 staining was initially graded independently by two investigators (I.K. and J.K.), blinded to other case information, into grades 0, 1+, 2+ and 3+ according to the ASCO/CAP 2018 guidelines, and a consensus grade was established together for discordant cases in a second round. HER2 staining heterogeneity was defined as sharply distinct tumor areas characterized by at a least two-degree difference in membrane staining intensity within the sample. The p53 staining was also first assessed independently by two investigators (I.K. and G.L.), blinded to other case information, then the discrepant cases were jointly assessed again in a second round. P53 status of a case was considered negative if the positive nuclear staining rate was between 1–49% of the total number of tumor cells (corresponding to wild type p53), while 0% and 50–100% staining rates were considered as positive p53 status (abnormal p53 IHC pattern) [19]. 

FISH analysis of the *HER2* gene and centromeric region of Chromosome 17 (Chr17) was performed using ZytoLight SPEC *HER2*/CEN 17 Dual Color Probe Kit (ZytoVision GmbH, Bremerhaven, Germany). The slides were examined with Leica DM RXA (Leica Microsystems, Wetzlar, Germany) epifluorescence microscope equipped with DAPI, Spectrum Green and Spectrum Orange filters (Vysis, Downers Grove, IL, USA). Images were taken using Leica DFC365 FX monochrome camera and Leica CW4000 FISH analysis software. In each case, the FISH signals were independently evaluated by two examiners (E.K. and I.K.) throughout the slides, followed by expert validation and a consensus result (G.L.), blinded to other case information. *HER2* and Chr17 signals were counted in at least 20–20 nuclei from two different areas exhibiting the most amplified and/or polysomic tumor cell population. Only cells containing at least one copy each of *HER2* and Chr17 were scored. *HER2* amplification assessment was performed based on the ASCO/CAP 2018 guidelines for breast cancer [23]. NMIBCs with a Chr17 signal/cell ratio of at least 2.25 in the whole tumor cell population counted were considered to exhibit polysomy 17, as previously described [24]. Bladder cancers with a Chr17 signal/cell ratio of at least 3.45 in the whole tumor cell population counted were considered to have high-polysomy for Chr17, as previously described [10]. To ensure that even a small but highly polysomic distinct tumor cell population could be identified, the Chr17 signal/cell ratio was recorded not only in the whole sample, but also separately in the polysomic cell population in each case (hereafter referred to as “highly polysomic cell population”).

### 2.3. Statistical Analysis

Statistical analysis and visualization were performed using the programming language R (version 4.0.3) [25]. Fisher exact test was used for the comparison of the clinical parameters, pathological features and results of the IHC and FISH analyses. TTP survival curves were estimated using the Kaplan–Meier method generated by using the R packages ggplot2, survival and survminer, followed by log-rank analysis to determine the difference between two groups. Patients who died of other causes prior to progression were censored at the time of death. Univariate and multivariable Cox regression prognostic analyses were performed using the R package survival, and Harrell c-statistic was defined to measure the predictive capacity. Internal validation was performed using bootstrap resampling process (validate.cph package in R), with 1000 repetitions to provide unbiased estimate of model performance. Two-sided *p* values lesser than 0.05 were considered statistically significant.

## 3. Results

### 3.1. Patient and Tumor Characteristics

The age distribution of the NMIBC patients was in accordance with the literature [26]. Patient and tumor characteristics are detailed in Table 1 and Appendix A. 

### 3.2. Chr17 Polysomy, HER2 Expression and Gene Amplification Status of the NMIBCs

None of the control cases was positive for HER2, and all of them were non-amplified for *HER2* gene and non-polysomic for Chr17. The NMIBC cases with 3+ HER2 overexpression had significantly higher grade, stage and lower recurrence rate than the tumors with no or low HER2 expression. Heterogeneous HER2 expression was associated with higher grade. The seven *HER2*-amplified NMIBCs (all T1 and high-grade/G3) had significantly higher grade and stage than non-amplified cases. The 28 polysomic (including nine high-polysomy) cases had significantly higher stage, grade and progression rate than the non-polysomic tumors. These associations between clinicopathological characteristics, HER2 overexpression, *HER2* amplification and Chr17 polysomy status of the analyzed tumors are detailed in Figure 1, Table 2 and Appendix A. 

### 3.3. Potential Predictor Variables and Progression of NMIBCs

During the follow-up (range: 2–158 months; median: 77 months), progression to muscle-invasive disease was observed in 14 cases (15.56%). Results of the univariate and multivariate analysis are shown in Table 3, in Figure 2 and Appendix A. Neither the intensity nor the heterogeneity of the HER2 overexpression was associated with the progression in our cohort (Table 3, Appendix A). Conversely, *HER2* gene amplification, Chr17 polysomy, high-polysomy status or presence of distinct highly polysomic cell population were all associated with a significantly higher hazard ratio (HR) of progression (Table 3) and shorter TTP (Figure 2). In multivariate analyses, Chr17 polysomy, high-polysomy or a distinct highly polysomic cell population was found to be an independent prognostic factor according to both the WHO 1973 and WHO 2004/2016 grade classification systems (Table 3 and Appendix A). 

### 3.4. Addition of Chr17 Polysomy and HER2 Amplification Status Improves Accuracy of EAU and AUA Risk Stratifications

The distribution of the cases between risk groups according to the EAU, EORTC and AUA risk stratification systems is shown in Appendix A. Results of Cox regression analysis and Kaplan–Meier curves are depicted in Appendix A, Figure 2 and Appendix A.

Both the original and the updated (using either the 1973 or the 2004/2016 WHO grading) EAU risk categorization showed a significant positive correlation with HR of progression; however the concordance-indices of the updated EAU risk stratification were better for TTP prediction (c-index = 0.666 vs. 0.684/EAU WHO 1973/ and 0.741/EAU WHO 2004/2016/, respectively). No significant correlation was found between the progression and the EORTC risk stratification. The combination with either *HER2* gene amplification, Chr17 polysomy, Chr17 high-polysomy status or distinct highly polysomic cell population improved the overall predictive capacity of both the EAU WHO 1973 (from 0.684 to 0.699/0.784/0.725/0.771, respectively) and the EAU WHO 2004/2016 (from 0.741 to 0.755/0.793/0.789/0.795, respectively), as well as the AUA (from 0.728 to 0.745/0.786/0.781/0.773, respectively) risk stratification as calculated by the Harrell’s c-statistic (Appendix A). The addition of polysomy 17 to the EAU progression risk score increased the bootstrap-corrected c-indices, indicating the added prognostic value at internal validation as well (Appendix A).

We propose the use of Chr17 polysomy and *HER2* amplification status to screen for subsequently progressing NMIBCs, thereby stratifying the EAU 2004/2016 risk groups even more accurately (Figure 3), as discussed below.

### 3.5. Reclassification of G2 Tumors by Molecular Grading Could Improve the Accuracy of the EAU 2004/2016 Risk Stratification System

As a more precise EAU risk stratification depends on the use of the WHO 2004/2016 grading system, we next focused on the reclassification of the WHO 1973 G2 subgroup, which is clinically particularly relevant. 

TTP was not significantly different between high- and low-grade tumors of G2 NMIBCs, (*p* = 0.092, Appendix A), but it was significantly shorter in patients with Chr17 polysomic tumors (*p* < 0.001, Appendix A). Accordingly, we propose a molecular reclassification of the WHO grading of NMIBCs as follows: Chr17 polysomic G2 tumors are reclassified as high-grade and non-polysomic G2 tumors as low-grade, (G1 and G3 tumors remain low and high grade, respectively). Following this reclassification, high grade NMIBCs exhibited significantly shorter TTP than their low-grade counterparts (*p* < 0.001, Appendix A). Therefore, the updated EAU 2004/2016 risk score was also recalculated using this modified WHO grading of NMIBCs. This resulted in a significantly shorter TTP (*p* < 0.001) and higher HR of progression (HR = 12.070 (3.324–43.810), *p* < 0.001) for tumors in the high/very high EAU risk groups; furthermore, the new molecularly reclassified EAU 2004/2016 score had a higher c-index than with the conventional grading, indicating a better model (c-index of the modified EAU 2004/2016 = 0.796 vs. c-index of the original EAU 2004/2016 = 0.741).

### 3.6. Potential Applications of Chromosome 17/HER2 Copy Number Status in the Diagnostic Practice of Non-Muscle-Invasive Bladder Cancer to Improve the Prediction of Progression

We investigated whether Chr17 polysomy and *HER2* amplification status could be used in practice to screen for progressive cases and achieve a more accurate risk stratification by further classifying the EAU 2004/2016 risk groups. Among the high- and very high-risk cases, none of the non-*HER2* amplified, non-polysomic cases progressed, while the progression rate was 50% for polysomic and/or *HER2*-amplified tumors (Figure 4A). On the contrary, among low- and intermediate-risk NMIBCs, only one of the three progressing cases was Chr17 polysomic and/or *HER2*-amplified. Based on this, as significant added value of *HER2*/Chr17 ISH was obtained in the high- and very high-risk EAU categories, we propose to use it primarily in these risk groups and reclassify the non-*HER2* amplified, non-polysomic cases to the EAU 2 (intermediate) risk group. This approach can prevent unnecessarily strict follow-up and treatment schemes in about 40% of the high/very high-risk EAU cases (14 cases in this cohort) by predicting a lower-risk prognosis for these. As well as the overall accuracy rate for identifying the progressing cases as high/very high risk, NMIBC will be improved from 30% [11/36] to 50% [11/22]. 

Another possible strategy is to refine the G1, G2, and G3 categories of the WHO 1973 grading using Chr17 polysomy and *HER2* amplification status. Only one of the 20 G1 tumors progressed, and neither in this case, nor in the others, was polysomy 17 and/or *HER2* amplification present (Figure 4B). Among the G2 bladder cancers, 15 Chr17 polysomy cases (12 low polysomy [Chr17/cell ≥2.25 and <3.45] and 3 high polysomy [Chr17/cell > 3.45]) were observed, present in 6/7 progressing and 7/45 non-progressing cases. Of the 18 G3 tumors, six were progressive, all of which had high polysomy 17 and/or *HER2* amplification. Based on the above, we suggest that G1 tumors should continue to be considered as low risk (low grade/LG) and G3 tumors as high risk (high grade/HG) for progression. For G2 tumors, cases with no molecular abnormality are proposed to be considered at low risk of progression (LG), while tumors with either polysomy 17 or *HER2* amplification are at high risk of progression (HG). When EAU WHO 2004/2016 risk stratification is performed using this molecular method-assisted LG/HG grading approach, the accuracy of assigning progressive NMIBC cases to the EAU high/very high risk group improves markedly (from 30% [11/36] to 40% [11/27]), but when the additional three non-*HER2* amplified, non-polysomic cases are also reclassified to the intermediate risk category, the overall accuracy rate reaches 46% [11/24] (Figure 4C).

### 3.7. Correlation of p53 Protein Expression with Chromosome 17/HER2 Copy Number Status and Progression of Non-Muscle-Invasive Bladder Cancer 

P53 immunohistochemistry results were available in 85 out of 90 cases, as in five of the cases tumor staining was not possible due to technical reasons (e.g., tumor tissues were carved out during serial sectioning, etc.). *TP53* mutation-associated p53 protein expression pattern was not observed in any of the 12 control cases and was not significantly correlated with the progression of NMIBCs (univariate analysis, Table 3). However, statistically significant correlation was found between the p53 expression and Chr17 polysomy of NMIBCs (Appendix A). We also investigated the possible role of p53 expression in the differentiation between progressive and non-progressive cases in the EAU high-risk group. In the non-progressing EAU high-risk NMIBC subgroup, eight p53 positive and 16 negative cases were identified (Appendix A). On the contrary, five cases of the progressive EAU high-risk NMIBCs showed p53 positivity but the other five were negative. Although only polysomy 17/*HER2* amplified cases progressed into MIBC from the EAU high-risk NMIBC group (11 cases), a further 11 of the 25 non-progressing cases exhibited polysomy 17 and/or *HER2* amplification as well. Accordingly, we have investigated whether the p53 status would be applicable to select further non-progressing cases out of these polysomy 17 and/or *HER2* positive EAU high-risk NMIBCs. However, no such cut-off level of p53 positivity was found that can appropriately differentiate between the polysomy 17/*HER2* amplified and non-polysomic/non-amplified cases in the non-progressive subgroup of EAU high-risk NMIBCs.

## 4. Discussion

Clinicopathological factor-based progression scores of the EAU and AUA are used to stratify NMIBCs for disease management. The EAU scoring system is based on the EORTC progression score and has been recently updated with new clinical risk factors, incorporating both the WHO 1973 and 2004/2016 grading systems [4,6]. However, these risk assessment systems are far from accurate; for example, the 10-year progression rates for patients in the EAU high and very high-risk groups are only 14% and 53%, respectively [6]. In fact, some studies suggest that it may be even lower [27]. This implies that many patients are over-surveilled or even over-treated, so there is a great need to improve the accuracy of the progression risk assessment of NMIBCs. Here, we show that the proposed addition of Chr17 polysomy/*HER2* status-based classification to the EAU risk score of NMIBCs improves the accuracy of the prognostic stratification of these tumors. In particular, this approach is useful to reclassify the G2 tumors into LG and HG categories. By using this molecularly improved WHO 2004/2016 grading for the EAU progression risk score, a substantially better finding rate of progressive NMIBC cases can be achived in the EAU WHO 2004/2016 high/very high-risk categories (Figure 4). Thus, this molecular marker-assisted subclassification of patients can help to optimize disease management and follow-up strategies.

Our hypothesis was that some of the cytogenetic alterations’ characteristics for MIBC are already present in the NMIBC stage, which allows the prediction of cases progressing subsequently. This was confirmed in one of our previous studies in which we found that chromosomal copy number alterations, including Chr17 polysomy detected by the UroVysion FISH test from urine cytology samples, represent an independent prognostic factor for NMIBCs [10]. Therefore, we aimed to further investigate the prognostic value of Chr17 copy number alterations in NMIBC tissue samples, using the *HER2*/Chr17 dual in situ hybridization method routinely used in pathology diagnostics. Furthermore, since not only HER2 but also the p53 protein is encoded in chromosome 17, and both carry prognostic potential in bladder cancer, both are also frequently tested by immunohistochemistry in routine diagnostics; therefore, we also wanted to investigate whether they could provide added prognostic value to the copy number alterations of chromosome 17 and the updated EAU risk table [15,20]. 

As others reported, polysomy 17 is associated with higher grade and muscle invasive stages in bladder cancer [16,28], and the number of aneusomic cell populations is significantly higher in pT1 than in pTa tumors [29]. In line with these, we found that polysomy 17 is associated with higher grade and stage in NMIBCs and Chr17 copy number alterations are independent prognostic factors beyond the well-established clinical and histopathological risk factors. 

Although previous studies have reported a correlation between HER2 overexpression and higher stage and grade, we found no association between progression of NMIBCs and HER2 overexpression [12,13,14]. This might be due to different interpretations of HER2 IHC results (we considered only IHC 3+ cases as HER2-positive according to the ASCO/CAP 2018 breast cancer HER2 guideline, while others considered 2+ cases also positive or used other criteria), but is more likely explained by the wide variation in HER2 expression levels among Chr17 polysomic tumors [12,13,14,23]. Mohanty et al. described how the ASCO/CAP 2018 guideline reduces the HER2 positivity rate of the high-grade urothelial carcinomas compared to the 2013 version [30]. *HER2* amplification has also been described as being associated with higher stage, grade and poor prognosis in NMIBCs [15]. Our data show that *HER2*/Chr17 ISH performed according to the ASCO/CAP 2018 guideline is applicable in the risk assessment of NMIBCs, as it provides additional prognostic information for these patients.

Accordingly, we have demonstrated, the addition of Chr17 copy number alteration (polysomy, high-polysomy, highly polysomic distinct cell population in the sample) and *HER2* amplification status to the updated EAU risk assessment could further improve it with either the WHO 1973 or 2004/2016 grading system. Furthermore, we confirmed in our cohort that regardless of whether the WHO 1973 or 2004/2016 grading system is used, the updated EAU score outperforms the original version, but the WHO 2004/2016 grading results in a more accurate EAU risk stratification of NMIBCs. This can be further improved by molecular subclassification according to our proposal, as the absence of Chr17 polysomy and *HER2* amplification predicts a low risk of progression, and we therefore propose to reclassify the non-*HER2* amplified, non-polysomic cases from the EAU 2004/2016 high and very high-risk groups to the intermediate-risk group.

Aberrant nuclear expression of the p53 protein has been shown to be prognostic in high grade urothelial cancers [20]. Therefore, we hoped that by adding p53 expression data we could improve the accuracy of the progression risk estimation of NMIBCs, based on the EAU risk table and Chr17 polysomy/*HER2* amplification status. However, aberrant p53 expression did not show a significant association with progression of NMIBCs to MIBC in our study, despite the statistically significant correlation between p53 expression and Chr17 polysomy of NMIBCs. Moreover, expression status of p53 was also not effective enough to distinguish progressive from non-progressive cases in the EAU high-risk group, or in differentiating between polysomy 17/*HER2* amplified and non-polysomic/non-amplified cases in the non-progressive subgroup of EAU high-risk NMIBCs.

Although it can be challenging for the pathologist to decide whether G2 tumors are low- or high-grade, the reproducibility of the WHO 2004/2016 grading system is slightly better; however, this classification has not been shown to outperform the WHO 1973 classification in predicting disease recurrence and progression [31]. Especially in muscle-invasive tumors, which are almost exclusively high-grade (~95%), the prognostic value of the WHO 2004/2016 system is clearly limited. We found no significant difference in disease outcome between the WHO 2004/2016 low- and high-grade G2 tumors. However, molecular grading of G2 tumors into low- or high-grade subgroups (according to the absence or presence of Chr17 polysomy and/or *HER2* amplification) was able to separate them prognostically, resulting in a more precise risk stratification with the new EAU 2004/2016 score. Based on this, if the pathologist is uncertain about the WHO 2004/2016 grade, we recommend this kind of molecular grading of G2 tumors (see Figure 4), which can also help to reduce the interobserver variability.

In our cohort, the AUA scoring system provided the second-best performance after the EAU, but we found that its accuracy can also be further improved by addition of the Chr17 polysomy/*HER2* amplification status. Contrary to the EAU and AUA scores, the distinction between the low- and high-risk patients by the EORTC scoring system was not statistically significant in our study. This may be explained by the fact that the EORTC score was developed in the 1980s, whereas in this study the patients were included from the 2010s, when more advanced therapeutic options were already available [2,3,4].

One value of this study is the relatively higher progression rate of the NMIBCs compared to other studies (15% vs. 4.6–4.9%), which allowed us to evaluate relatively rare prognostic events in a cohort of only 90 patients [32,33]. Nevertheless, the cohort size is a limitation, which may be responsible for failure to identify some well-known prognostic factors such as tumor stage or EORTC score. The proportion of the female patients is higher in our study (48.9%), as would be expected from a consecutive cohort. Female gender is a prognostic factor for T1G3 tumors; however, this is unlikely to have influenced our results, as several studies have shown that gender is not a predictor of progression of NMIBCs into muscle-invasive disease [6,34]. Another important value of this study is that *HER2*/Chr17 ISH is one of the most widely available molecular pathology tests that does not require fresh–frozen tissue and can therefore be performed on routine FFPE histology specimens. Thus, the implementation of Chr17 polysomy/*HER2* amplification testing would provide an immediate and simple solution to further refine the prognostic risk assessment of NMIBCs in the uro-oncology practice. However, due to the retrospective nature of this study, treatment regimens for patients were not standardized, and neither regular surveillance for early detection of cancer recurrence and progression nor uniform follow-up periods were established. In view of the aforementioned details, further multicenter external validation of the independent prognostic factors identified in the present single center study in additional retrospective and prospective cohorts is needed.

## 5. Conclusions

In summary, our study confirmed the improved risk assessment of the updated EAU risk score compared to the original version. Our findings clearly show the prognostic potential of in situ hybridization-based determination of Chr 17 polysomy and *HER2* amplification in bladder cancer, while the detection of HER2 and p53 protein expression by immunohistochemistry can neither approximate the prognostic stratification results of the genetic-based method nor further increase its accuracy. The addition of chromosome 17 polysomy/*HER2* amplification status to the updated EAU and AUA scores improves their accuracy and identifies subsets of EAU high-risk NMIBC patients with a modest- and very high risk of progression. Using this approach for molecular grading of G2 tumors also improves the accuracy of risk stratification by EAU score. Thus, for the WHO 1973 histopathological grading of bladder cancers, we suggest a molecular reclassification of the G2 NMIBCs as follows: we propose to classify Chr17 polysomic and/or *HER2* amplified G2 tumors as high-grade (HG) and non-polysomic G2 tumors as low-grade (LG) NMIBCs (G1 and G3 tumors remain graded as low- and high-grade, respectively). Furthermore, as an introduction of our results into the routine practice of risk assessment of NMIBC patients, we propose to reclassify non-*HER2*-amplified, non-polysomic NMIBCs from the EAU 2004/2016 high- and very high-risk groups (categorized by either conventional histological grading or molecularly assisted grading) to the intermediate-risk group using the *HER2*/Chr17 ISH. Although we concluded that these molecular markers are suitable for implementation in clinicopathological factor-based progression risk stratification and tumor grading of non-muscle-invasive bladder cancer, confirmatory validation in larger cohorts would be desirable before widespread use. However, the application of this approach in routine practice may be facilitated by the fact that *HER2*/Chr17 in situ hybridization is the same technique that is widely used in the everyday pathological workflow of breast cancer diagnostics; it can therefore be easily and quickly adapted to bladder cancer diagnostics. On the other hand, it is a relatively inexpensive molecular test (about 80–250 EUR/case, depending on the manufacturer and the additional costs), and an automated version of which adapted for immunohistochemistry staining systems is also available in many pathology departments. Its use can therefore be cost-effective, considering that by excluding those NMIBC cases from the high- and very high-EAU risk categories that do not require extra follow-up and medical care, it will be possible to reduce the burden on both patients and the healthcare system.

## Figures and Tables

**Figure 1 cancers-14-04570-f001:**
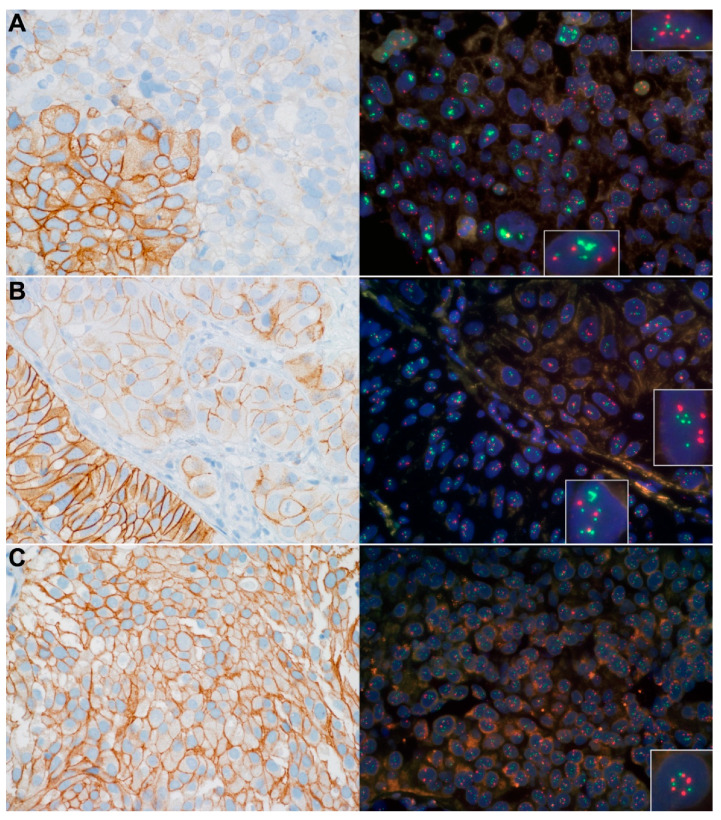
HER2 protein immunohistochemistry, *HER2* gene/Chromosome 17 centromere dual color fluorescence in situ hybridization (FISH) test results from three different cases (case **A**–**C**). *HER2* amplification was mostly (except in one case) associated with strong HER2 protein expression. Chr17 polysomy was found to be associated with a broad spectrum of HER2 immunohistochemical expression, ranging from negative to strong (3+) positivity. Positivity of HER2 IHC (on the left) appeared as brown cell membrane staining (original magnification: 600×). *HER2*/Chr17 FISH (on the right; typical cell nuclei are shown at higher magnification in insets) signals appeared as follows: *HER2* gene—green dots and/or clusters of dots; Chromosome 17 centromere—red dots; Nucleus—blue color (original magnification: 630×). (**A**): Pronounced HER2 protein expression heterogeneity: strong complete membrane positivity (3+) in the left-lower quadrant and weak (0–1+) HER2 staining in the remaining part. Polysomy 17 with associated *HER2* gene amplification (*HER2*/Chr17 ratio ≥ 2) (lower inset) was detected in the area of 3+ HER2 IHC positivity while only IHC 0–1+ expression was associated with the high polysomy of Chr17 (upper right inset). The histological phenotype of *HER2* amplified and non-amplified tumor parts was similar. (**B**): HER2 protein expression heterogeneity: strong complete membrane positivity (3+) on the left-lower area and moderate (2+) HER2 staining in the remaining part. Polysomy 17, with associated *HER2* gene amplification (*HER2*/Chr17 ratio ≥ 2) (lower inset), was detected in the area of 3+ HER2 IHC positivity, while only IHC 2+ expression was associated with the polysomy of Chr17 (upper right inset). The histological phenotypes of *HER2* amplified and non-amplified tumor parts were markedly different. (**C**): Diffuse HER2 protein expression: Strong complete membrane staining in the whole tumor cell population (3+). High polysomy of Chr17 without amplification of the *HER2* gene. IHC: immunohistochemistry; Chr17: Chromosome 17.

**Figure 2 cancers-14-04570-f002:**
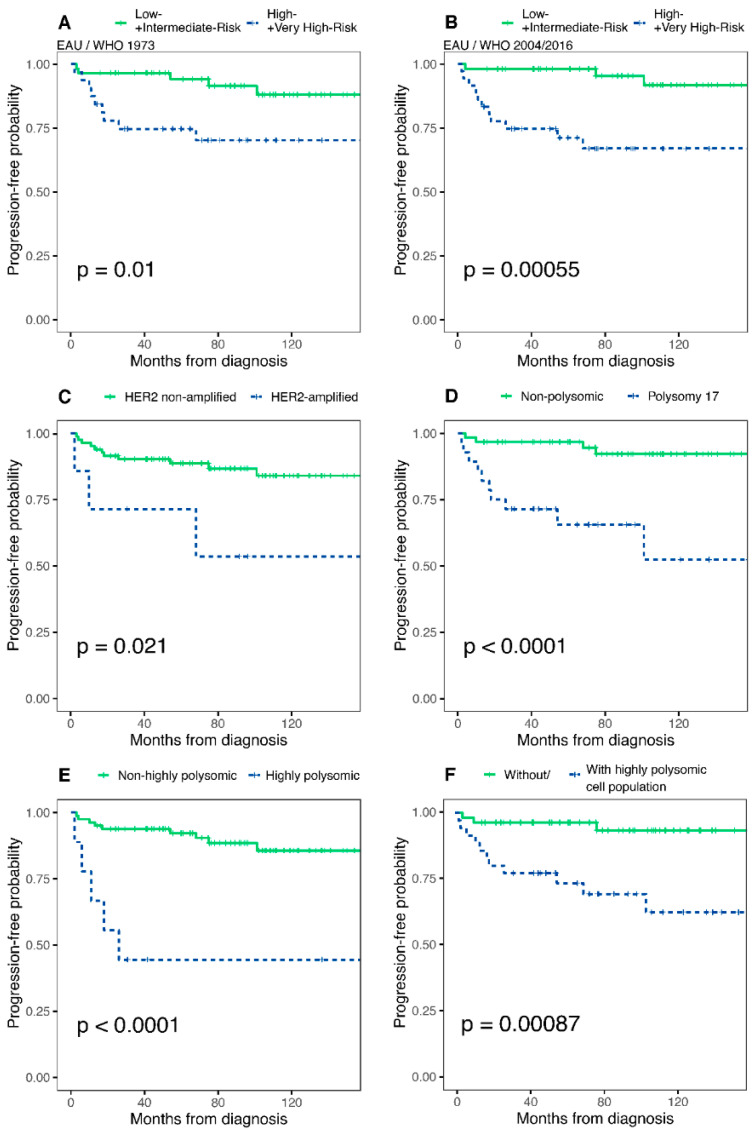
Time-to-progression curves in all patients for (**A**) EAU 1973 low+intermediate vs. high+very high-risk, (**B**) EAU 2004/2016 low+intermediate vs. high+very high-risk, (**C**) *HER2* gene amplification status, (**D**) Chromosome 17 polysomy status, (**E**) Chromosome 17 high polysomy status, (**F**) Presence of distinct highly polysomic cell population. Progressive disease is defined as progression to stage T2 or higher stage disease. *p*-values (log-rank test) are indicated in each figure. EAU: European Association of Urology.

**Figure 3 cancers-14-04570-f003:**
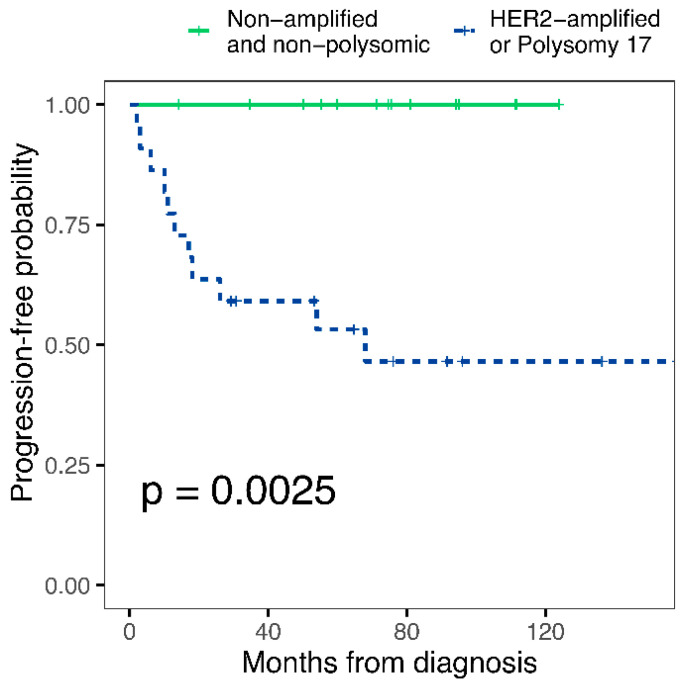
Time-to-progression curves of EAU 2004/2016 high+very high-risk patients for Chr17 polysomy and *HER2* amplification. Progressive disease is defined as progression to stage T2 or higher stage disease. *p*-value (log-rank test) is indicated in the figure. EAU: European Association of Urology; Chr17: Chromosome 17.

**Figure 4 cancers-14-04570-f004:**
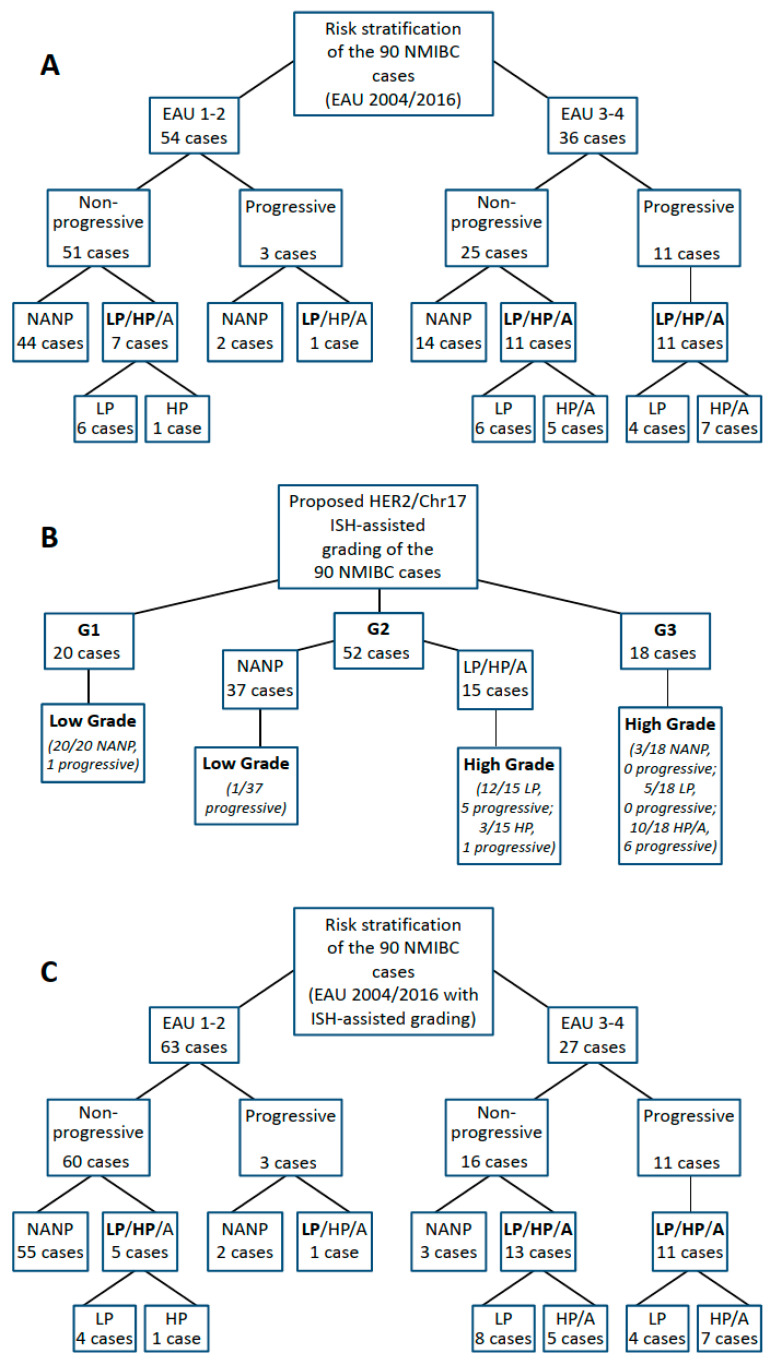
Potential applications of chromosome 17/*HER2* copy number status in the diagnostic practice of non-muscle-invasive bladder cancer to improve the prediction of progression. (**A**): Progression rates in the EAU 2004/2016 risk groups. Among the high- and very high-risk cases (EAU 3–4 risk groups), none of the non-*HER2* amplified, non-polysomic cases progressed, while the progression rate was 50% for polysomic and/or *HER2*-amplified tumors. Therefore, we propose to reclassify the non-*HER2* amplified, non-polysomic EAU 3–4 cases to the EAU 2 (intermediate) risk group to prevent unnecessarily strict follow-up and treatment for these patients. By this, the overall accuracy rate for identifying the progressing cases as high-/very high-risk NMIBC will be improved from 30% [11/36] to 50% [11/22]. (**B**): Another possible diagnostic use of the Chr17 polysomy and *HER2* amplification status is to refine the G1, G2, G3 categories of the WHO 1973 grading. We suggest that G1 NMIBCs should continue to be considered as low risk (low grade/LG) and G3 tumors as high risk (high grade/HG) for progression. Regarding G2 tumors, non-*HER2* amplified, non-polysomic cases are proposed to be considered at low risk of progression (LG), while tumors with either polysomy 17 or *HER2* amplification are at high risk of progression (HG). (**C**): When EAU WHO 2004/2016 risk stratification is performed using this molecular method-assisted LG/HG grading approach (discussed above, in the **B**), the accuracy of assigning progressive NMIBC cases to the EAU high/very high-risk group improves markedly (from 30% [11/36] to 40% [11/27]), but when the additional three non-*HER2* amplified, non-polysomic cases are also reclassified to the intermediate-risk category, the overall accuracy rate reaches 46% [11/24]. Bold letters indicate which molecular alteration is present in the given subgroup. EAU: European Association of Urology, NMIBC: non-muscle-invasive bladder cancer, NANP: non-*HER2* amplified and non-polysomic for chromosome 17, LP: chromosome 17 polysomy, HP: chromosome 17 high polysomy, A: *HER2* amplification, WHO: World Health Organization, ISH: in situ hybridization.

**Table 1 cancers-14-04570-t001:** Patient and tumor characteristics.

* Patient characteristics *		
**Age at diagnosis**	**mean**	**range**
	65.84	(40–91)
**Gender**	**n**	**%**
Male	46	(51.11)
Female	44	(48.89)
**Months of follow up**	**median**	**range**
	77	(2–158)
**Intravesical instillation (BCG/Chemo ever)**	**n**	**%**
Yes	61	(67.78)
No	22	(24.44)
unknown	7	(7.78)
* Tumor characteristics *	*n*	*%*
**Tumor type**		
Primary	70	(77.78)
Recurrent	20	(22.22)
**Stage**		
pTa	42	(46.67)
pT1	47	(52.22)
pTis	1	(1.11)
**Grade**		
1	20	(22.22)
2	52	(57.78)
3	18	(20.00)
Low grade	47	(52.22)
High grade	43	(47.78)
**Multiplicity**		
Solitary	79	(87.78)
Multiple	11	(12.22)
**Tumor size**		
<3 cm	68	(75.56)
≥3 cm	22	(24.44)
**Recurrence**		
Yes	54	(60.00)
No	36	(40.00)
**Progression to T2**		
No	76	(84.44)
Yes	14	(15.56)

BCG: Bacillus Calmette-Guerin.

**Table 2 cancers-14-04570-t002:** Relation of the *HER2*/Chr17 fluorescence in situ hybridization results and the clinicopathological characteristics of the tumors.

	*HER2* Amplified		*HER2* Non-Amplified		*p*	Chr17 Polysomic		Chr17 Non-Polysomic		*p*	Chr17 High Polysomic		Chr17 Non-High Polysomic		*p*
* Tumor characteristics *	n	%	n	%		n	%	n	%		n	%	n	%	
**Tumor type**															
Primary	5	(71.43)	65	(78.31)		23	(82.14)	47	(75.81)		8	(88.89)	19	(23.46)	
Recurrent	2	(28.57)	18	(21.69)	0.649	5	(17.86)	15	(24.19)	0.592	1	(11.11)	62	(76.54)	**<0.001**
		CI of OR		0.103–7.878			CI of OR		0.434–5.787			CI of OR		3.047–1168.346	
**Stage ***															
pTa	0	(0.00)	42	(50.60)		6	(21.43)	36	(58.06)		1	(11.11)	41	(50.62)	
pT1	7	(100.00)	40	(48.19)	**0.013**	21	(75.00)	26	(41.94)	**0.002**	7	(77.78)	40	(49.38)	0.062
pTis	0	(0.00)	1	(1.20)		1	(3.57)	0	(0.00)		1	(11.11)	0	(0.00)	
		CI of OR		0.000–0.718			CI of OR		0.061–0.634			CI of OR		0.003- 1.185	
**Grade ****															
1	0	(0.00)	20	(24.10)		0	(0.00)	20	(32.26)		0	(0.00)	20	(24.69)	
2	0	(0.00)	52	(62.65)	**<0.001**	15	(53.57)	37	(59.68)	**<0.001**	3	(33.33)	49	(60.49)	**0.002**
3	7	(100.00)	11	(13.25)		13	(46.43)	5	(8.06)		6	(66.67)	12	(14.81)	
		CI of OR		0.000–0.129			CI of OR		0.025–0.371			CI of OR		0.013–0.491	
Low grade	0	(0.00)	47	(56.63)		4	(14.29)	43	(69.35)		0	(0.00)	47	(58.02)	
High grade	7	(100.00)	36	(43.37)	**0.004**	24	(85.71)	19	(30.65)	**<0.001**	9	(100.00)	34	(41.98)	**<0.001**
		CI of OR		0.000–0.579			CI of OR		0.017–0.263			CI of OR		0.000–0.402	
**Tumor size**															
<3 cm	5	(71.43)	63	(75.90)		20	(71.43)	48	(77.42)		5	(55.56)	63	(77.78)	
≥3 cm	2	(28.57)	20	(24.10)	1	8	(28.57)	14	(22.58)	0.600	4	(44.44)	18	(22.22)	0.214
		CI of OR		0.119–8.965			CI of OR		0.240–2.346			CI of OR		0.070–2.022	
**Multiplicity**															
Solitary	7	(100.00)	72	(86.75)		27	(96.43)	52	(83.87)		9	(100.00)	70	(86.42)	
Multiple	0	(0.00)	11	(13.25)	0.591	1	(3.57)	10	(16.13)	0.162	0	(0.00)	11	(13.58)	0.594
		CI of OR		0.190-infinity			CI of OR		0.665–233.295			CI of OR		0.264-infinity	
**Recurrence**															
Yes	3	(42.86)	51	(61.45)		15	(53.57)	39	(62.90)		5	(55.56)	49	(60.49)	
No	4	(57.14)	32	(38.55)	0.431	13	(46.43)	23	(37.10)	0.487	4	(44.44)	32	(39.51)	1
		CI of OR		0.065–3.004			CI of OR		0.251–1.864			CI of OR		0.162–4.449	
**Progression**															
Yes	3	(42.86)	11	(13.25)		10	(35.71)	4	(6.45)		5	(55.56)	9	(11.11)	
No	4	(57.14)	72	(86.75)	0.073	18	(64.29)	58	(93.55)	**<0.001**	4	(44.44)	72	(88.89)	**0.004**
		CI of OR		0.618–32.735			CI of OR		1.968–38.464			CI of OR		1.724–58.357	

* Ta vs. T1 tumors; ** Grade 1/2 vs. Grade 3; Statistically significant *p* values are displayed in bold; FISH: fluorescence in situ hybridization; Chr17: chromosome 17. The statistical analyses shown in the table were carried out using Fisher’s exact test.

**Table 3 cancers-14-04570-t003:** Univariate and multivariable Cox regression analysis of potential predictor variables and time-to-progression.

Variable	Category	HR	95% CI	*p*
*TOTAL cohort—Univariate analyses*				
Age, years	continuous variable	1.032	(0.982–1.086)	0.216
Tumor type	Recurrent vs. primary (Ref.)	2.691	(0.934–7.760)	0.067
T stage	Tis, T1 vs. Ta (Ref.)	3.354	(0.921–12.210)	0.066
Histologic grade (WHO 1973)	Grade 3 vs. grade 1–2 (Ref.)	3.619	(1.248–10.490)	**0.018**
Histologic grade (WHO 2004/2016)	High grade vs. low grade (Ref.)	5.243	(1.441–19.070)	**0.012**
Tumor size	≥3 cm vs. <3 cm (Ref.)	0.791	(0.221–2.839)	0.719
Tumor multiplicity	Multiple vs. solitary (Ref.)	<0.01	(0.000-infinity)	0.998
HER2 expression	3+ vs. 1/2+ or 0 (Ref.)	1.727	(0.386–7.727)	0.475
Heterogeneity for HER2 expression	Heterogenous vs. non-heterogenous (Ref.)	0.667	(0.223–1.992)	0.468
*HER2* gene amplification	Amplified vs. non-amplified (Ref.)	4.036	(1.122–14.520)	**0.033**
Chromosome 17 polysomy	≥2.25 vs. <2.25 signal/cell (Ref.)	7.440	(2.306–24.000)	**<0.001**
Chromosome 17 high polysomy	≥3.45 vs. <3.45 signal/cell (Ref.)	7.505	(2.478–22.730)	**<0.001**
Highly polysomic cell population	Yes vs. No (Ref.)	6.577	(1.832–23.610)	**0.004**
P53 IHC status	1–49% vs. 0% and 50–100%	2.427	(0.7826–7.524)	0.125
*MODEL 1—Multivariable analysis*				**0.010**
Age, years	continuous variable	1.03462	(0.980–1.092)	0.217
Histologic grade (WHO 2004/2016)	High grade vs. low grade (Ref.)	4.27592	(1.111–16.457)	**0.035**
*HER2* gene amplification	Amplified vs. non-amplified (Ref.)	2.37480	(0.615–9.174)	0.210
*MODEL 2—Multivariable analysis*				**0.001**
Age, years	continuous variable	1.042	(0.983–1.105)	0.169
Histologic grade (WHO 2004/2016)	High grade vs. low grade (Ref.)	2.411	(0.574–10.121)	0.229
Chromosome 17 polysomy	≥2.25 vs. <2.25 signal/cell (Ref.)	5.139	(1.391–18.983)	**0.014**
*MODEL 3—Multivariable analysis*				**0.003**
Age, years	continuous variable	1.02938	(0.971–1.091)	0.331
Histologic grade (WHO 2004/2016)	High grade vs. low grade (Ref.)	3.35917	(0.828–13.632)	0.090
Chromosome 17 high polysomy	≥3.45 vs. <3.45 signal/cell (Ref.)	4.01119	(1.206–13.339)	**0.024**
*MODEL 4—Multivariable analysis*				**0.001**
Age, years	continuous variable	1.051	(0.991–1.114)	0.096
Histologic grade (WHO 2004/2016)	High grade vs. low grade (Ref.)	2.944	(0.766–11.322)	0.116
Highly polysomic cell population	Yes vs. No (Ref.)	5.403	(1.004–12.259)	**0.015**

Statistically significant *p* values are displayed in bold; HR: hazard ratio; CI: confidence interval.

## Data Availability

All data generated or analyzed during this study are included in this published article and its Appendix A or available following reasonable requests from other authors or investigators.

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
