# Peer review of "Addition of Chromosome 17 Polysomy and HER2 Amplification Status Improves the Accuracy of Clinicopathological Factor-Based Progression Risk Stratification and Tumor Grading of Non-Muscle-Invasive Bladder Cancer"

_cancers, 2022, doi:10.3390/cancers14194570_

Round 1

Reviewer 1 Report (Previous Reviewer 2)

I read the revised version of the work:" Addition of chromosome 17 polysomy and HER2 amplification 1 status improves the accuracy of clinicopathological factor-based progression risk stratification and tumor grading of non-muscle-invasive bladder cancer".  The nomenclature of the spelling of the names of genes is still not properly used. The authors write about genes, and the spelling of the gene name corresponds to the spelling for proteins. I also have a few questions:

1) how the authors explain the finding of a high rate of TP53 protein overexpression (16/90) in the non-advanced stages of bladder cancer. This marker is detected mainly in the higher stages of the disease or the carcinoma  in situ (one case in that study).

2) why the authors took the 50% limit as the equivalent of overexpression for the TP53 protein

3) what about the cases where there is no polysomy of chromosome 17 and the expression of TP53 proteins is reduced, because the gene TP53 is a tumor suppressor gene

4) I miss a final and clear conclusion on what we should use in a routine application: IHC or FISH assessment, or both. This is important because it is two different tests, longer time and of course cost and whether we should abandon the determination of TP53 protein expression, as well as how it relates to the molecular subclassification of bladder cancers

Author Response

Comments and Suggestions for Authors

I read the revised version of the work:" Addition of chromosome 17 polysomy and HER2 amplification 1 status improves the accuracy of clinicopathological factor-based progression risk stratification and tumor grading of non-muscle-invasive bladder cancer".  The nomenclature of the spelling of the names of genes is still not properly used. The authors write about genes, and the spelling of the gene name corresponds to the spelling for proteins. I also have a few questions:

Thank you for the helpful comments.
Regarding the gene and protein nomenclature, we agree with the reviewer that there is a need to properly distinguish gene and protein symbols. The TP53 gene and p53 protein symbols are established and conventional notations, but to make the distinction even clearer, we have italicized TP53 in accordance with the conventional notation for gene symbols (now TP53 and p53). The HER2 symbol is widely used for the ERBB2 gene and its product. To be consistent, we use capital italics to denote the gene, while the protein is written in regular capital letters (now HER2 and HER2).

1) how the authors explain the finding of a high rate of TP53 protein overexpression (16/90) in the non-advanced stages of bladder cancer. This marker is detected mainly in the higher stages of the disease or the carcinoma  in situ (one case in that study).

In order to better demonstrate the relationship between p53 expression and the other variables examined, we have included a further detailed table (Supplementary Table 6). This shows that of the 85 cases examined by p53 immunohistochemistry, a total of 14 showed p53 overexpression (16.5%). We agree with the reviewer that p53 expression is higher in more advanced stage and higher grade bladder cancers, as our own data in Supplementary Table 6 show. Accordingly, p53 overexpression was found in 4/41 pTa and 9/43 pT1 stage bladder cancers (and the only carcinoma in situ), 1/44 low grade (LG) and 13/41 high grade (HG) tumors. Note that the fact that the proportion of cases with p53 overexpression is higher in more advanced bladder cancers may also be due to the fact that the proportion of HG cases increases significantly with higher pT stage. As is well known, the proportion of LG tumors among muscle-invasive bladder cancers (≥pT2) is only around 1% (Morera et al. Clinical Parameters Outperform Molecular Subtypes for Predicting Outcome in Bladder Cancer: Results from Multiple Cohorts, Including TCGA. J Urol. 2020;203(1):62-72. doi:10.1097/JU.0000000000000351). In a study by Hodgson et al. among muscle-invasive high-grade bladder cancers, the proportion of cases expressing p53 in 50-100% of tumor cells was 45% (Hodgson et al. p53 immunohistochemistry in high-grade urothelial carcinoma of the bladder is prognostically significant. Histopathology. 2017;71(2):296-304. doi:10.1111/his.13225). Therefore, we do not consider 16.5% to be a high rate in our overall cohort, even among our high-grade NMIBC cases, only 32% of cases showed overexpression of p53 (in 50-100% of tumor cells).

2) why the authors took the 50% limit as the equivalent of overexpression for the TP53 protein

We adapted the approach of Hodgson et al. on the p53 positivity cut-off levels, as this group developed criteria for p53 positivity specifically for bladder cancer (Hodgson A et al. Reassessment of p53 immunohistochemistry thresholds in invasive high grade bladder cancer shows a better correlation with TP53 and FGFR3 mutations. Pathol Res Pract. 2020;216(11):153186. doi:10.1016/j.prp.2020.153186). In essence, the immunohistochemical approach that best correlates with TP53 mutation status in bladder cancer was defined, resulting in p53 positive being defined as the absence of immunohistochemical staining (less than 1% of positive nuclei) and overexpression, with a cut-off level of 50% or more of positive nuclei. Accordingly, a positive nuclei rate between 1 and 49% is considered as immunohistochemical p53 negativity in bladder cancer, as it is described in the Methods section (2.2. Immunohistochemistry (IHC) and fluorescence in situ hybridization (FISH) analysis, lines 143-146).

3) what about the cases where there is no polysomy of chromosome 17 and the expression of TP53 proteins is reduced, because the gene TP53 is a tumor suppressor gene

As shown in Supplementary Table 6, of the 56 non-progressive, non-polysomy 17 and non-HER2 amplified NMIBC cases examined by p53 immunohistochemistry, 3 had a p53 positive cell percentage below 1%, 3 had a percentage ≥50% and 50 cases had a percentage between 1-49%. Of the 4 progressive non-polysomy 17 and non-HER2 amplified NMIBC cases, all 4 were p53 immunohistochemically negative (p53 positive nuclei between 1-49%). Thus, unfortunately, neither reduced nor elevated p53 expression is able to select progressive cases among non-polysomy 17 and non-HER2 amplified NMIBC tumors.

4) I miss a final and clear conclusion on what we should use in a routine application: IHC or FISH assessment, or both. This is important because it is two different tests, longer time and of course cost and whether we should abandon the determination of TP53 protein expression, as well as how it relates to the molecular subclassification of bladder cancers

Thank you for drawing our attention to the need to better emphasize the main messages of our study in the manuscript. Our results unambiguously demonstrate that determination of Chr 17 polysomy and HER2 amplification by in situ hybridization is of prognostic significance in bladder cancer, while neither immunohistochemical detection of HER2 nor p53 protein expression can approximate the prognostic stratification ability of the genetic method nor further increase its accuracy. As molecular subtyping is mainly based on RNA expression results, and we know (and our own results for protein expression support this) that there can be significant differences between DNA level abnormalities and their expression at RNA and protein levels, we did not seek a relationship with molecular subtyping in the present study. As previously reported (Kocsmár I et al. Development and Initial Testing of a Modified UroVysion-Based Fluorescence In Situ Hybridization Score for Prediction of Progression in Bladder Cancer. Am J Clin Pathol. 2020;153(2):274-284. doi:10.1093/ajcp/aqz165), the potential prognostic significance of chromosomal copy number changes was raised by the diagnostic application of the UroVysion Bladder Cancer Detection Kit based on fluorescence in situ hybridization detection of tumor cells voided in urine. This kit can detect the vast majority of bladder cancers regardless of molecular classification, indicating that some common cytogenetic abnormalities may occur across different RNA-based molecular subtypes. As our practical experience has shown that chromosomal aneusomies on chromosomes 3, 7 and 17, which are part of cytogenetic abnormalities that accumulate during progression and are detected by UroVysion, almost always occur simultaneously. Thus, chromosome 3 and 7 copy number alterations without aneusomy 17 are virtually absent, and we hypothesized that HER2/ chromosome 17 ISH, which is widely used in breast pathology, could provide prognostic results, which our results confirmed. We ourselves were surprised to find that neither HER2 nor p53 immunohistochemical approaches were able to further increase the prognostic accuracy of the ISH, but this also highlights the potential discrepancy between protein expression results and DNA level changes.

For a final and clear conclusion, as requested by the reviewer, the final conclusion was completed as follows:

„Our findings clearly show the prognostic potential of in situ hybridization-based determination of Chr 17 polysomy and HER2 amplification in bladder cancer, while the detection of HER2 and p53 protein expression by immunohistochemistry can neither approximate the prognostic stratification results of the genetic-based method nor further increase its accuracy.”

“Accordingly, as an introduction of our results into the routine practice of risk assessment of NMIBC patients, we propose to reclassify non-HER2-amplified, non-polysomic NMIBCs from the EAU 2004/2016 high- and very high-risk groups to the intermediate-risk group using the HER2/Chr17 ISH. As well as for the WHO 1973 histopathological grading of bladder cancers, we suggest a molecular reclassification of the G2 NMIBCs as follows: we propose to classify Chr17 polysomic and/or HER2 amplified G2 tumors as high-grade (HG) and non-polysomic G2 tumors as low-grade (LG) NMIBCs (G1 and G3 tumors remain graded as low- and high-grade, respectively).”

Reviewer 2 Report (New Reviewer)

1.      Was multiplicity of tumors not a variable in the Cox regression?

2.      Some of the known prognostic factors did not emerge as significant in the cox regression-can authors discuss- was administration of BCG significantly prognostic? If BCG was not prognostic, it questions the validity of analysis.

3.      Can authors present at least an internal validation since there is no external validation presented.

4.      The limitations need to be better discussed. The authors do highlight some differences in this dataset from real world datasets, but a multicenter external validation could be proposed.

Author Response

Comments and Suggestions for Authors

  1. Was multiplicity of tumors not a variable in the Cox regression?

Thank you for noting this. Multiplicity was included in the Cox regression analysis as one of the variables and is now shown in the Table 3. Since the univariate analysis showed no significant difference between multiple and single tumors, this variable was not included in the multivariable model.

  1. Some of the known prognostic factors did not emerge as significant in the cox regression-can authors discuss- was administration of BCG significantly prognostic? If BCG was not prognostic, it questions the validity of analysis.

Thank you for bringing this up. Detection of statistically significant prognostic factors is strongly influenced by both the size of the cohort and its composition and design. Accordingly, a sufficient number of patients with a given prognostic factor should be included in the study, and on the other hand, it should be ensured that an equally adequate number of control patients are also enrolled. In the case of BCG treatment, this would mean that the cohort would have to ensure not only the inclusion of a sufficient number of patients with standardized BCG treatment in subgroups defined by known or presumed prognostic factors, but also the enrolment of a group of patients who are homogeneous in all respects and have not been treated. As the cohort was not designed in this way, and the patients who were eligible for BCG treatment according to the current treatment guidelines based on the known evidence of efficacy of BCG have been treated, no untreated control group was available. However, this also ensures that those treated with BCG are not expected to be significantly different prognostically from those not receiving BCG treatment, so that the homogeneity of the cohort with respect to the prognostic factors we examined is better ensured than if a control population eligible for but not receiving BCG treatment had been included.

  1. Can authors present at least an internal validation since there is no external validation presented.

Acknowledging the importance of the reviewer's point, internal validation was performed using bootstrap resampling process with 1000 repetitions to provide unbiased estimate of model performance. The bootstrap-corrected c-indices are shown in Supplementary Table 4. Basically, similar c-indices were achieved at internal validation in comparison with the original c-indices, addition of polysomy 17 to the EAU score still increased the c-indexes, indicating the added prognostic value.

Accordingly, the 3.4. of the Results section was supplemented by the next sentence:

“Addition of polysomy 17 to the EAU progression risk score increased the bootstrap-corrected c-indices, indicating the added prognostic value at internal validation as well (Supplementary Table 4).”

  1. The limitations need to be better discussed. The authors do highlight some differences in this dataset from real world datasets, but a multicenter external validation could be proposed.

In agreement with the reviewer's request, we have added further study limitations and addressed the need for multicenter external validation as follows:

“However, due to the retrospective nature of this study, treatment regimens for patients were not standardized, and neither regular surveillance for early detection of cancer recurrence and progression nor uniform follow-up periods were established. In view of the aforementioned details, further multicentre external validation of the independent prognostic factors identified in the present single centre study in additional retrospective and prospective cohorts is needed.”

Round 2

Reviewer 1 Report (Previous Reviewer 2)

The authors made the necessary corrections.

This manuscript is a resubmission of an earlier submission. The following is a list of the peer review reports and author responses from that submission.

Round 1

Reviewer 1 Report

Authors performed an interesting analyses testing the effect of HER2 expression and heterogeneity on survival in patients with NMIBC. The manuscript is scientifically sound, but still I have serious concerns.

  1. It is not clear how in the same model could be included boht HER2 expression and heterogeneity (table 2). Such covariates seems to be highly collinear in my opinion
  2. How many readers assessed HER2 expression? Is there high concordance between them? If no concordance was reached how it was handled?
  3. Those pathologist and HIC readers were blinded on regard the first diagnosis? How many pathologists were involved?
  4. Several models have been proposed in table 2. No clue about covariates selection process have been provided. Please clarify in methods section
  5. In those with a progression which was the status at the second TURB? Did the expression status evolved in time?
  6. How many surgical procedure were performed for each patients?
  7. Did authors examined the association between recurrence and progression?
  8. Confidence intervals for C-index and other metrics should be provided. No internal validation (i.e. bootstrap or cross validation) have been reported
  9. Please provide the power of included analyses.

Reviewer 2 Report

The markers mentioned in the study have already been tested many times and their role in the etiopathogenesis of bladder cancer is known. The work is prepared very meticulously, although it does not add much to what is already known. Several points need to be clarified:

1)Who or what biological samples was included in the control group and what results were obtained

2) please provide complete follow-up data (a 2-month follow-up period is not really an observation period that would allow any conclusions to be drawn with the scheme applicable to control cystoscopy)

3)the summary does not explain how these markers change the evaluation, there is no information as to how in routine practice these analyzes should be performed, as well as whether it is possible to perform them easily and quickly and how expensive they are

4)Table 2 does not provide information on which test was performed for each calculation and how, for example, significance was obtained for chromosome 17 high polysomic and chromosome 17 non-high polysomic in the group with recurrent cancer

5)On the same chromosome 17 there is a locus for the TP53 gene, but FISH assessment has been performed only for HER2 gene

6) the conclusions: "these molecular markers may be suitable for implementation in clinicopathological factor-based progression risk stratification and tumor grading of non-muscle-invasive bladder cancer, after further validation in larger cohorts." are less certain than the summary :", the addition of Chr17 polysomy and HER2 amplification status to the EAU risk table improves its accuracy in predicting disease progression"